# Insights on Molecular Characteristics of Hydrochars by ^13^C-NMR and Off-Line TMAH-GC/MS and Assessment of Their Potential Use as Plant Growth Promoters

**DOI:** 10.3390/molecules26041026

**Published:** 2021-02-15

**Authors:** Laís G. Fregolente, João Vitor dos Santos, Giovanni Vinci, Alessandro Piccolo, Altair B. Moreira, Odair P. Ferreira, Márcia C. Bisinoti, Riccardo Spaccini

**Affiliations:** 1Laboratório de Estudos em Ciências Ambientais, Campus São José do Rio Preto, Instituto de Biociências, Universidade Estadual Paulista (Unesp), Letras e Ciências Exatas, Rua Cristovão Colombo 2265, Jardim Nazareth, São José do Rio Preto 15054-000, Brazil; laisfregolente@hotmail.com (L.G.F.); joaovitor_piaca12@hotmail.com (J.V.d.S.); altair@ibilce.unesp.br (A.B.M.); marcia.bisinoti@unesp.br (M.C.B.); 2Interdepartmental Research Centre on Nuclear Magnetic Resonance (NMR) for the Environment, Agro-Food and New Materials (CERMANU), Università degli Studi di Napoli Federico II, Via Università 100, 80055 Portici, Italy; giovanni.vinci@unina.it (G.V.); alessandro.piccolo@unina.it (A.P.); 3Laboratório de Materiais Funcionais Avançados (LaMFA), Universidade Federal do Ceará, Fortaleza 60455-900, Brazil; opferreira@fisica.ufc.br

**Keywords:** hydrothermal carbonization, sugarcane residues, molecular characteristics, water-soluble fractions, plant growth promoter

## Abstract

Hydrochar is a carbon-based material that can be used as soil amendment. Since the physical-chemical properties of hydrochar are mainly assigned to process parameters, we aimed at evaluating the organic fraction of different hydrochars through ^13^C-NMR and off-line TMAH-GC/MS. Four hydrochars produced with sugarcane bagasse, vinasse and sulfuric or phosphoric acids were analyzed to elucidate the main molecular features. Germination and initial growth of maize seedlings were assessed using hydrochar water-soluble fraction to evaluate their potential use as growth promoters. The hydrochars prepared with phosphoric acid showed larger amounts of bioavailable lignin-derived structures. Although no differences were shown about the percentage of maize seeds germination, the hydrochar produced with phosphoric acid promoted a better seedling growth. For this sample, the greatest relative percentage of benzene derivatives and phenolic compounds were associated to hormone-like effects, responsible for stimulating shoot and root elongation. The reactions parameters proved to be determinant for the organic composition of hydrochar, exerting a strict influence on molecular features and plant growth response.

## 1. Introduction

The use of biomass as feedstock to produce new carbon-based materials has been described as valuable recycling method based on two main techniques, hydrothermal carbonization (HTC) and pyrolysis [1,2,3]. These processes allow the use of different biomasses producing carbonaceous materials with different physical and chemical properties. The solid material from pyrolysis is called biochar while the solid product generated in HTC is defined as hydrochar. The HTC process also generates a liquid product named process water, which has been used as fertilizer [1,4] or recycled in new HTC process (recirculation) [5,6,7]. The solid products from HTC and pyrolysis can be destined for different purposes as solid fuel [8,9], adsorbents [10], catalysis [11], batteries and supercapacitors [12], among others [2,13]. Biomass recycled by HTC and pyrolysis have also showed up reliable properties as new alternative options to dispose agriculture residues through their carbonization that generate chars with valuable potential to be applied as soil conditioners [14,15,16,17,18]. 

The use of carbonaceous materials as soil amendment has been reported to improve crop development by indirect effects on soil properties, such as nutrients availability to plant uptake, increase of cation exchange and water holding capacities, and promote the carbon storage [19,20,21,22], as well as by direct stimulation of plant growth [22,23,24]. However, the characteristics of carbonaceous materials are closely linked to the process parameters, including the kind of feedstock, that affect the effectiveness of agricultural application [17,21,25]. Recent studies have highlighted a correlation between the effects on plants growth (or better seed germination) and the presence of bioavailable phenolic compounds, in dissolved organic fractions derived either from lignocellulosic biomass or charred-material [22,23,24,26,27]. Moreover, the observed increase of either plant development or resistance to biotic stress [28,29], promoted by bio- and hydro-char application have been associated to the triggering of hormone-like activity [22,27,30].

On the other hand, also negative or undesirable side effects have been ascertained upon hydrochar and biochar soil application [31,32], associated to the presence of either phytotoxic organic compounds or to their inorganic composition [32,33]. Since a defined correspondence between the molecular features of complex organic matter and plant cellular pathways has not yet been fully clarified, the evaluation of reaction conditions on the composition of hydrochars may be a helpful requirement to predict its effects on crop development. 

Both ^13^C cross-polarization magic angle spinning nuclear magnetic resonance spectroscopy (^13^C-CPMAS-NMR) and off-line pyrolysis with tetramethylammonium hydroxide (thermally assisted hydrolysis and methylation) followed by gas chromatography-mass spectrometry (off-line TMAH-GC/MS), are updated and powerful tools for the analytical investigation of both recycled biomasses and derived fractions with limited sample pre-treatments [22,23,24]. The non-destructive solid state ^13^C-NMR technique is an acknowledged reference methodology for the rapid elucidation of overall carbons distribution between the main classes of organic compounds in complex matrices. The combination of ^13^C-CPMAS-NMR with the thermally assisted hydrolysis and methylation methodology may improve the analytical appraisal of composition, transformation, and reactivity of natural organic matter (NOM), with a suitable integration and refining of the intrinsic low molecular resolution of solid-state NMR. In fact, the preparative off-line pyrolysis followed by the GC–MS allows the use of large quantities of sample, providing the release and reliable qualitative identification and quantitative estimation of specific components and biomarkers, thereby being recognized as valuable technique for structural investigation and to trace both origin and modification or recycled biomasses. 

The hydrothermal carbonization of agro-industrial wastes and residues stand out as an environmentally friendly process, since mild temperatures are generally applied, and the starting biomass can be dry or wet [34]. The use of sugarcane bagasse or vinasse, or the mixture of both, has been described as an efficient way to treat this industry by-products, obtaining hydrochar with high carbon, nutrient content and potential release of volatile compounds [35,36,37]. Hydrochars produced with phosphoric acid as additive showed a higher nutrient immobilization. On the other hand, the use of sulfuric acid in the reaction medium leaded to a hydrochar with a higher carbon content [35,36]. These characteristics would be interesting to hydrochar soil application, probably, being able to improve soil fertility or soil carbon storage, respectively.

However, most of the studies with hydrochar from sugarcane industry by-products are focused on hydrochar characteristics as morphology, structure, and inorganic composition. Furthermore, scarce information is still available on the detailed feature of organic matter composition of these materials and the structural-activity relationship with respect to their effects on plant development. Here the organic composition of different hydrochars were investigated more deeply through ^13^C-NMR and off-line TMAH-GC/MS. This study enabled the identification of how reaction parameters changed the hydrochar organic matter, allowing to predict which hydrochar can be used to agronomic purposes based on its composition through a direct evaluation by plant response. So, this study aimed (i) to investigate the organic composition of different hydrochars, to elucidate their molecular properties; and (ii) assess the relationship between hydrochars composition and seed development through germination bioassay, using hydrochar water-soluble extracts, to evaluate hydrochars as potential plant growth promoters. 

## 2. Results and Discussion

### 2.1. Hydrochar Characterization

In respect to inorganic composition, a larger incorporation of nutrient occurred in the hydrochar produced with vinasse rather than process water (Figure 1). This finding has to be related to initial vinasse composition characterized by a higher concentration of nutrients than process water [4,37]. Moreover, following previous report [36,37] the comparison between the vinasse-based hydrochars shows that a more effective immobilization of nutrients is achieved when phosphoric acid is employed in the reaction. The use of vinasse also favored the increase of Mg, Cu, Mn, and Zn content in hydrochar. 

The molecular composition of hydrochars was investigated using the solid-state ^13^C-NMR spectra (Figure 2) and the off-line TMAH-GC/MS. According to the chemical shifts, the NMR spectra were assigned into seven main regions, representing the main organic functional groups: alkyl C (0–43 ppm); methoxyl substituent on aromatic ring or carbon bound to nitrogen derivatives, OCH_3_/NCH (43–60 ppm); alkyl carbon bonded to one oxygen atom, i.e., *O*-alkyl (60–90 ppm); alkyl carbon bonded to two oxygen atoms, O-C-O anomeric (90–110 ppm); proton- and alkyl-substituted aromatic carbon, aromatic C-C/C-H (110–145 ppm); oxygen-substituted aromatic carbon, aromatic C-O (145–160 ppm) and ester, carboxyl and amide carbon, COO/N-C=O (160–190 ppm) [38,39,40,41,42]. 

The NMR spectra showed the presence of nonpolar aliphatic and aromatic groups as predominant components of the chemical structures of hydrochars, thus revealing a prevalent hydrophobic character. The signals at 14 ppm and 30 ppm (Figure 2) found in all samples, correspond to methyl and methylene carbons structures, respectively, typical of aliphatic linear lipid compounds, e.g., fatty acids, waxes, bio-polyesters [27,42,43]. The peaks centered at 54 ppm may be assigned to either lignin-derived structures or to peptide moieties [42,43,44]. The remarkable signal in all hydrochars at around 128 ppm was due to the large incorporation of aromatic carbons, while the shoulders, at 144 ppm indicated the inclusion lignin and phenolic unite. The low intensity on NMR resonances in the 160–190 ppm chemical shift region suggests an intense decarboxylation, leading to the almost complete removal of carboxylic functional groups.

The estimation of relative carbons distribution and the corresponding structural indexes of hydrochar samples are shown in Table 1. The integration of NMR spectra showed that aromatic C-C/C-H and alkyl-C, accounting for the 28.3–43.2% and the 27.7–42.9% of total area, in the order, were the dominant structures. The lower amount of carbohydrates in the O-alkyl region and the absence of COO/N-C=O from peptides and proteins showed that hydrolysis reactions took place during the thermal treatment, turning the bioavailable polar materials into more aromatic and apolar alkyl carbon compounds. 

The structural features of hydrochars stemmed by the thermal decarboxylation and dehydration of polar groups and leading to higher hydrophobic character, were elucidated by the determination of structural index (Table 1). The larger values found for *HB*/*HI* parameter confirmed the significant shared predominance of apolar fractions in all samples, with differential contribution of aliphatic and aromatic compounds. In fact, the *ARM* index showed that carbonization process in the presence of process water produced a larger conversion in aromatic rings in respect to vinasse-based hydrothermal medium (Table 1). Conversely, the *A*/*AO* ratios suggested the prevalence of alkyl-C chains in vinasses derived hydrochars and a larger residual presence of hydrophilic structures of polysaccharides in the R-Ph and R-Su samples [27,45]. The close correspondence between the intensities of methoxyl-lignin substituent at 54 ppm with those of O-aryl C functions, highlighted by the low level of Lignin ratios in all samples (Table 1), further support the lack of bioavailable peptide moieties in the 43–60 ppm region and the contribution of phenolic derivatives to the signals of 145–160 interval [23,45]. The HTC reactions with sulfuric acid, promoted the formation of the most hydrophobic hydrochars with more aromatic moieties and larger apolar aliphatic molecules in combination with process water and vinasse, respectively (Table 1). Based on the acknowledged relation between hydrophobic components and high stability level of organic matter, this finding could be an indication that the sulfuric acid promotes the biomass conversion of polar functional groups into more recalcitrant and less decomposable organic materials [36]. 

The TMAH-thermochemolysis of hydrochars released around 167 compounds (Figure 3 and Appendix A) identified as methyl ether and ester of alkyl and aromatic monomers grouped into main organic classes and quantified in terms of relative yield (Figure 4). The conversion of cellulose components of sugarcane residues by hydrothermal carbonization process was confirmed by the small amount of carbohydrates derivatives identified in the pyrograms. This result has to be also related to the analytical limits of the thermochemolysis technique. In fact, a poor efficiency of off-line thermochemolysis to detect carbohydrates and polysaccharides in complex matrices is attributed to the thermal behavior and rearrangement of polyhydroxy compounds during pyrolysis that affect the suitable release of polysaccharides [23,27,45].

The most abundant thermochemolysis compounds were represented by benzene derivatives (BD) and aliphatic esters (ES) followed by phenolic compounds (Figure 4), thereby supporting the C distribution found in NMR analysis. The large fraction of benzene derivatives originated from lignin units and lignin by-products of the carbonization reactions, and they were identified as hydroxyphenyl, guaiacyl, and syringyl monomers [23,45]. In line with NMR data the R-Su samples released the larger yield of aliphatic compounds and the lower relative amount of phenolic derivatives, while the higher relative percentage of BD and phenolic molecules were found in hydrochar samples added with phosphoric acid (Figure 4). The relative lower content of aromatic compounds found in the TMAH-GC/MS for Su and R-Su as compared to NMR analyses may be explained by the strong reaction condition brought by sulfuric acid on hydrothermal medium which contribute to increase the aliphatic, and aromatic hydrophobic character in Su and R-Su samples respectively, evidenced by NMR index. The intense acid hydrolytic potential may have thus promoted an intense transformation of initial lignocellulose biomass in both aliphatic chains with a concomitant improved condensation of newly formed aromatic compounds gathered in poly-condensed materials. The relative mild analytical temperatures of off-line pyrolysis reaction were not able to breakdown the aromatic ring, thereby lowering the final yield of BD monomers.

### 2.2. Bioassay

The seeds germination with water soluble fractions of different hydrochars did not show neither a delay in germination rates (Appendix A) nor significant statistically differences among the treatments and control sample (Appendix A), thus indicating that the germination process was not negatively influenced by the concentration and chemical composition of water-soluble fractions. 

A significant correlation between seed gemination and dissolved TOC amounts was found for the treatments with water soluble extracts of Ph and R-Ph samples (Appendix A). A positive dose response effect with increasing germination percentage at larger TOC concentration was shown for Ph treatment. The germination index (*GI*), which is linked to the speed of germination, suggested a potential biostimulation of germination response that was improved with the concentration of water-soluble TOC levels of Ph hydrochar. Conversely, an opposite trend, i.e., a decrease in *GI* with the increase of water-soluble OC concentration, was shown by the R-Ph sample, while the application of dissolved fraction from hydrochar added with sulfuric acids, Su and R-Su samples, did not reveal a specific variation in the *GI* pattern (Appendix A). 

Figure 5 shows the root (Figure 5a) and shoot (Figure 5b) length measured after 7 days of seed germination. The different soluble fractions had a shared positive influence on maize growth, stimulating the seedlings development. Notwithstanding the restrained statistical significances, the raising in TOC concentration of water-soluble fraction improved the root development in all treatments specially at C_100_ treatment for Su (*p* = 0.0007), R-Su (*p* = 0.001), and R-Ph (*p* = 0.02) samples. The dissolved fraction from vinasse-based hydrochar added with phosphoric acid showed the best significant performances at all tested concentrations with larger effect for C50 (*p* = 0.0001) and C100 (*p* = 0.0002) (Figure 5a).

The analysis of shoot development showed a similar interaction with the applied OC solutions, being the soluble fraction of Ph sample (Figure 5b), the unique application with positive statistical differences (*p* < 0.05) for C_10_ (*p* = 0.005), C_50_ (*p* = 0.001), and C_100_ (*p* = 0.003) assays. Although the other samples promoted the shoot growth with the incremental TOC doses of water-soluble fraction, no statistical significance was achieved among them. 

Although the biological activity of dissolved organic fractions was thoroughly investigated in recent years, no univocal structural activity relationships have been highlighted to explain the main mechanisms of action [46,47]. The most acknowledged hypotheses identify in the combination of both hydrophobic structural properties and specific molecular bioactive components the main variables that trigger the potential bio-effector properties of various organic extracts [24,44]. The wrapping envelope formed by hydrophobic components may create a micelle-like or microemulsion carrying systems that allow a feasible physical and biochemical preservation of bioactive fragments thereby promoting a favorable adhesion to roots tissues followed by a structural rearrangement and release of bioactive molecules [48,49]. As pointed out in previous studies on bioactive dissolved organic materials from recycled biomasses, the different responses in seedling development can be attributed to the molecular composition inherited from corresponding bulk biomass composition, determined by NMR and TMAH-GC/MS analyses [23,27,44,45,46].

The hydrochar samples from HTC reactions with phosphoric acid showed the most effective promoting activity on maize root and shoot elongation (Figure 5). The combination of hydrophobic characteristics and the larger content of bioactive lignin fragments and phenolic derivatives, highlighted by both NMR and thermochemolysis, may be hence associated to the assumed bio-stimulant effect of Ph treatment.

The release of bioavailable phenolic and lignin derivatives compounds (145–160 ppm) from dissolved organic materials have been linked to promoting of root and shoot elongation [23,45,46]. This behavior is described as a hormone-like effect, which outcome is similar to biochemical pathways regulated alternatively by either gibberellin and mainly auxin activity or to stress mediated reactive molecules such as ROS [47,50,51]. 

For the process water- and sulfuric acid-based hydrochars samples, the structural features identified by NMR stressed the preservation of polysaccharides components in R-Ph and R-Su, joined for Su and R-Su to a lower TMAH-GC/MS detection of benzene-lignin moieties and phenolic derivatives. It may hence conceivable that the addition of process water and sulfuric acid may have favored either the incorporation of less altered lignocellulose fragments from bagasse and vinasse residues or the more intense condensation of aromatic and apolar aliphatic components in bulk hydrochars. This composition may have reduced the potential release of bioactive compounds in the water-soluble fractions. The data of root and shoot elongations revealed a significant effect only at largest OC addition for R-Ph, Su and R-Su (Figure 5). This behavior may be associated to the release of suitable amounts of soluble bioactive compounds only at larger OC concentration as well as to the folding of rigid hydrophobic components on root surfaces that may produce a plant metabolic and physiological responses activated by the so-called transient beneficial mild-stress or eustress [47]. 

Besides the organic composition, the hydrochar inorganic composition could also interfere on seeds development. The higher concentration of nutrients in water-soluble fraction could positively affect the seedling development. On the other hand, the larger salt content may also increase the electrical conductivity and the osmotic potential of the growing medium that could reduce the seeds development, since it could limit the root uptake capacity. Furthermore, the presence of Al, K, Na, Ca, and Fe at high concentrations are reported as potential inhibitors of seed germination, which negatively influence the root elongation [52,53]. On the other hand, it has been also pointed out that a moderate osmotic stress can activate a hormone signaling pathways related to the increase of auxin transport and consequently both H+-ATPase activity and root elongation [47]. These elements are present in all analyzed hydrochars and K, Mg, Ca, and Fe showed the highest concentrations in the Ph sample (Figure 1), whose dissolved fraction promoted the best seed development. However, previous studies revealed that K, Ca, and Mg in hydrochar are present as insoluble inorganic phases [36,37], as phosphates, being not released during water extraction. 

Based on the present results probably the organic composition of hydrochars seems to overcome these potential adverse effects of nutrient or osmotic characteristics of inorganics components in Ph and R-Ph, and also in Su and R-Su samples, since no negative effects were identified. Nevertheless, the development of adaptation mechanisms by the seedling to the stress conditions should be also considered. The absence, or the low concentration of organic components able to stimulate maize seedling growth in Su and R-Su samples is an indicative that the organic composition exerts greater influence on root and shoot development than the inorganic composition.

## 3. Materials and Methods

### 3.1. Hydrothermal Carbonization Process

The choice of HTC parameters, the use of sugarcane bagasse, vinasse or process water, and phosphoric acid or sulfuric acid was made considering the previous studies aiming hydrochar soil application [36,37,54,55,56] (the sugarcane bagasse collection was authorized and registered with the National System for the Management of Genetic Heritage and Associated Traditional Knowledge (SisGen) N°A0018C2).

So, the hydrochars were obtained from hydrothermal carbonization of liquid vinasse and sugarcane bagasse mixture (Table 2, reactions 1 and 2), or process water and sugarcane bagasse mixture (Table 2, reactions 3 and 4), in the presence of sulfuric or phosphoric acids. The biomass/liquid ratio was 1:20 *w*/*v*, and it was used phosphoric acid (4% *v*/*v*) or sulfuric acid (4% *v*/*v*) as additives [36,37]. The hydrochar samples are hence divided into two groups, one carbonized with phosphoric acid (Ph and R-Ph samples) and the other with sulfuric acid (Su and R-Su samples). 

The reactions were carried out in a handmade stainless-steel coated Teflon^®^ closed reactor (600 mL) the temperature of carbonization was 230 ± 10 °C, time reaction was 13 h, and the pressure was self-generated. After reaction, the reactor was coaled in an ice bath. The hydrochar was separated from the process water by reduced pressure filtration, washed with deionized water until constant pH, and dried in an oven at 50 °C for 24 h. The description of each reaction parameter is reported in Table 2. 

### 3.2. Hydrochars Nutrients Content

For quantification of the nutrients in the hydrochars, the desired sample were submitted to acidic decomposition with nitric acid and hydrogen peroxide following the 3050B method [57]. The concentration of total Mn, Al, Ca, K, Mg, Fe, Cu, and Zn were determined by Flame Atomic Absorption Spectrometry (FAAS) (Varian, model AA240FS). Potassium was determined in the emission mode.

### 3.3. Off-Line TMAH-Thermochemolysis Coupled with Gas Chromatography and Mass Spectrometry (GC-MS)

The thermal treatment was performed applying 200.0 mg of hydrochars. The samples were placed in a quartz cuvette, adding 500 µL of tetramethylammonium hydroxide solution (25% *v*/*v* in methanol), and air-dried for 2 h. Then the sample was put in a Pyrex tubular reactor (model F21100) (Barnstead International, Dubuque, IA, USA), heated at 400 °C for 10 min, with a heating rate of 20 °C min^−1^, and under He flow. The components obtained during pyrolysis were continually transferred to two successive chloroform traps, kept in an ice bath. The solutions were combined and concentrated using a vacuum system. The combined extracts were dried under a stream of N_2_ and dissolved in 100 µL of dichloromethane. The pyrolytic process was performed in duplicate for each sample.

#### GC/MS Parameters

Analyses were performed using a gas chromatographer (PerkinElmer, model Auto System XL, CA, USA) and a RTX-5MS WCOT capillary column (Restek, 30 m × 0.25 mm; film thickness 0.25 µm) coupled to a quadrupole mass spectrometer (PerkinElmer, model Turbo PE Turbomass-Gold, San Jose, CA, USA). Helium was used as a carrier gas at 1.00 mL L^−1^, in the split mode with a split ratio of 1:30. Mass spectra were obtained in El mode (70 eV) and scanned from *m*/*z* 45 to 650. The sample injection was done in an initial oven temperature of 60 °C for 1 min, raised to 100 °C at 7 °C min^−1^ and then to 300 °C at a rate of 4 °C min^−1^, and held for 10 min. The GC-MS analyze was made in duplicate for each sample, and the compound identification by chromatograms were made according to the NIST database.

### 3.4. Solid-state ^13^C Nuclear Magnetic Resonance

The solid-state ^13^C nuclear magnetic resonance (NMR) was carried out by CPMAS NMR (Bruker, model AV 300-MH, Billerica, MA, USA) equipped with a cross polarization/magic angle spinning probe with a bore of 4.0 mm. A 4.0 mm diameter zircon rotor was packaged with ~80 mg of sample. The carbon spectra were acquired from a rotation frequency of 10 kHz, with a repetition time of 1 s, a contact time of 1 ms, an acquisition time of 20 ms and 4000 scans.

The relative intensities of each C-containing structure were calculated by integrating ^13^C-NMR peak areas for specific regions, such as, alkyl C (0–43 ppm); OCH_3_/NCH (43–60 ppm); O-alkyl (60–90 ppm), O-C-O anomeric (90–110 ppm), aromatic C-C/C-H (110–145 ppm), aromatic C-O (145–160 ppm), and COO/N-C=O (160–190 ppm). The structural differences of hydrochars may be highlighted by dimensionless structural indexes determined from the relative amount of C distribution in the NMR spectra [23,27]. The hydrophobicity index (*HB*/*HI*) is obtained by the integrated relative areas of hydrophobic aliphatic and aromatic carbons over those of hydrophilic carbons (Equation (1)). The aromatic index (*ARM*) is the ratio of aryl components over alkyl compounds, calculated following Equation (2). The alkyl ratio (*A*/*AO*) compares the signal area of apolar aliphatic molecules with the polar alkyl carbons (Equation (3)). The lignin ratio (*LR*) correlates the intensities of methoxyl lignin substituents and C-N bonds to the relative amount of O-aryl-C and phenolic compounds (Equation (4)).

The first three parameters have been extensively applied to estimate the biochemical stability and molecular hydrophobicity of organic materials, while the *LR* is a useful indicator to discriminate between signals owing to lignin and other phenolic moieties (lower *LR*) with respect to the prevalent inclusion of peptide moieties in the 45–60 ppm interval (larger *LR*).
(1)$$HB/HI=\left[\frac{\sum \left[\left(0-43\mathrm{ppm}\right)+\left(43–60\right)/2+\left(110-160\mathrm{ppm}\right)\right]}{\sum \left[\left(43–60\right)/2+\left(60–110\mathrm{ppm}\right)+\left(160–190\;\mathrm{ppm}\right)\right]}\right]$$
(2)$$ARM=\left[\frac{\left(110-160\right)\mathrm{ppm}}{\sum \left(0-43\mathrm{ppm}\right)+\left(60-110\mathrm{ppm}\right)}\right]$$
(3)$$A/AO=\;\left[\frac{\left(0-43\mathrm{ppm}\right)}{\left(60-110\mathrm{ppm}\right)}\right]$$
(4)$$LR=\;\left[\frac{\left(0-43\mathrm{ppm}\right)}{\left(145-160\mathrm{ppm}\right)}\right]$$

### 3.5. Germination Assay

The water-soluble fractions from hydrochars were obtained at room temperature with deionized water as extractor using the proportion of 1:15 *w*/*v*. The samples were stirred for 12 h, and then separated by centrifugation (3500 rpm for 15 min) [44,58]. The aqueous phase was filtered twice, applying filter paper. Hydrochar soluble fractions dilutions were carried out based on carbon concentration, using concentrations of 0, 1, 10, 50 and 100 mg C L^−1^ (named as C_0_, C_1_, C_10_, C_50_, and C_100_, respectively) for germination studies.

The germination experiments were carried out as described by Fregolente et al. [4], using commercial maize seeds (*Zea mays*) (Seminis, 85% germination). Ten seeds were used for each Petri dish (15 cm diameter), with five replicates for each concentration, placed in a germination B.O.D. chamber (MA 403, Marconi), maintained at 25 °C for 7 days and a lighting schedule of 16 h light and 8 h dark. The experiments were monitored daily and when a length ≥ 0.5 mm of shoot or root emerged from the seed it was considered germinated. The experiment lasted 7 days, and in the end the total number of germinated seeds was recorded, and the germination index was calculated using the Equation (5) [4,59,60]. The root and shoot lengths were measured using the ImageJ software (version 1.51i).
(5)$$GI=\sum \frac{ni}{ti}$$
where *ni* = number of seeds germinated on the observation day and *ti* = observation day.

### 3.6. Statistical Analysis

The normality of the data was evaluated using normal probability plots of residuals, and the comparison between groups was performed using a general linear model (GLM). Tukey’s post-hoc test was performed when results were considered statistically significant, for a *p* value < 0.05.

## 4. Conclusions

The different process parameters in HTC conversion of residual biomasses, such as the inorganic acids used as reactions additives and the replacement of vinasse by process water have shown to be determinant to define the molecular properties of hydrochars. The NMR spectroscopy indicated that all hydrochars have mainly hydrophobic characteristics. However, the use of phosphoric acid in the reaction medium promoted an incorporation of more bioavailable benzene derivatives pertaining to lignin structures and phenolic compounds, while sulfuric acid produced a more intense modification of initial biomass in largely hydrophobic hydrochar with organic composition based on aliphatic and condensed aromatic structures. 

The results from maize seedling assays enabled a correlation with the identified molecular composition. The differences on the stimulation of root and shoot length were attributed to the hormone-like activity, which was probably exerted by lignin-benzene derivatives and phenolic compounds identified in hydrochars. These compounds are present in different concentrations in hydrochar samples, showing higher content in Ph and R-Ph hydrochars. Ph hydrochar was the only one able to promote both maize root and shoot elongation, mainly at C_50_ and C_100_ organic carbon concentration. The results further confirm the validity of molecular characterization of recycled organic biomass as useful requirement to support the more suitable agronomic applications. The preliminary analyses on molecular characteristics of hydrochars provide valuable information for a comprehensive understanding of potential bioactive properties and suitable application of bulk samples and derived fractions. In this context the HTC conversion of residual sugarcane biomasses may hence act as a viable complementary source to obtain valuable organic products, thus contributing to implement the virtuous NOM recycle and support the sustainable SOM management and improve the circular economy in agro-ecosystems.

## Figures and Tables

**Figure 1 molecules-26-01026-f001:**
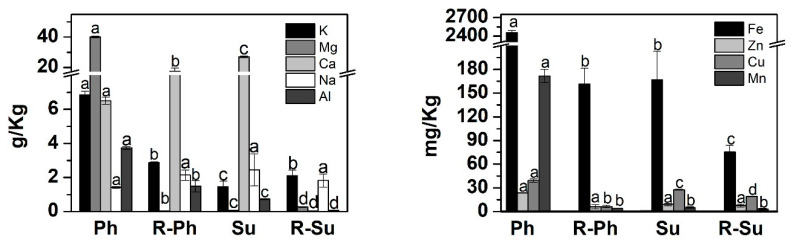
Potassium (K), magnesium (Mg), calcium (Ca), sodium (Na), aluminium (Al), iron (Fe), zinc (Zn), copper (Cu), and manganese (Mn) contents in hydrochar samples (Ph, R-Ph, Su, and R-Su). The letters “a”,”b”, “c”, and “d” were used to indicate statistical difference. Same letters on bars indicate non-significant differences between treatments, and same letters indicate significant differences between treatments. The error bars represent standard deviation (SD).

**Figure 2 molecules-26-01026-f002:**
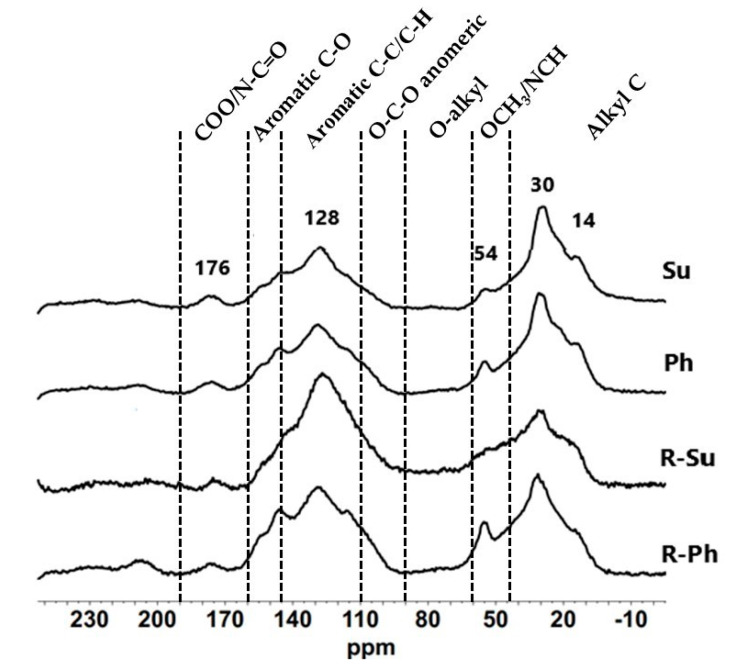
Solid-state ^13^C-NMR spectra of hydrochars (Su, Ph, R-Su, and R-Ph).

**Figure 3 molecules-26-01026-f003:**
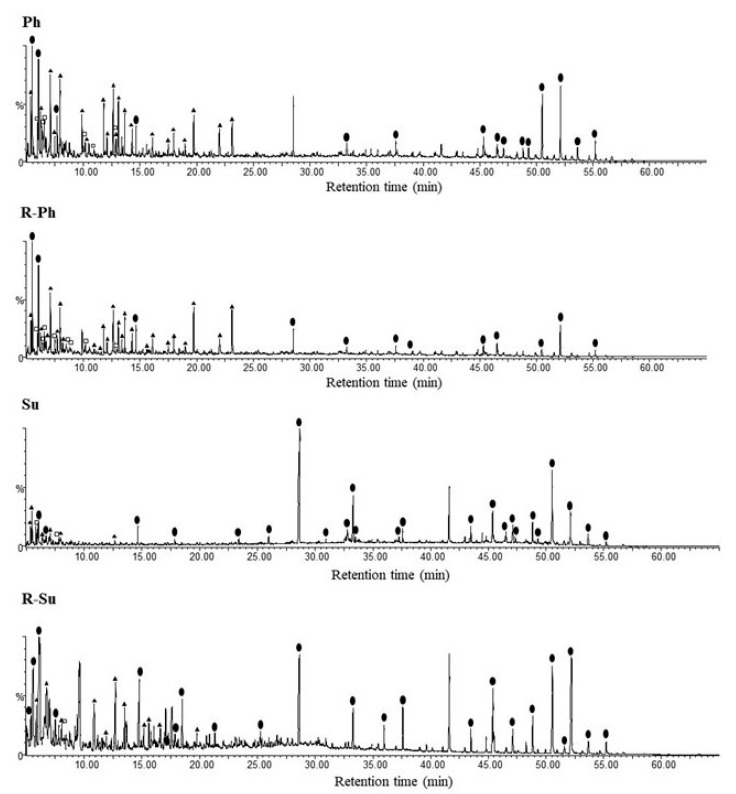
Chromatograms of thermochemolysis products of the four hydrochars (Ph, R-Ph, Su, and R-Su). (□) phenolic compounds; (▲) benzene derivative; (●) aliphatic esters.

**Figure 4 molecules-26-01026-f004:**
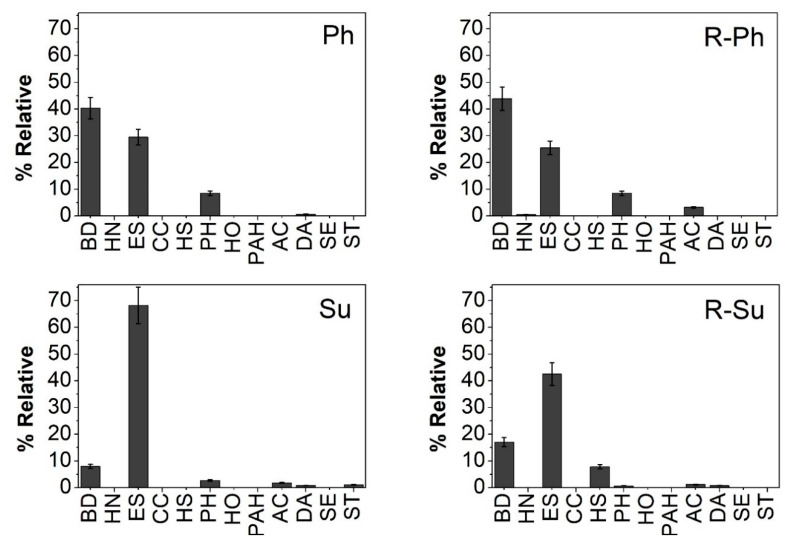
Relative percentage of main classes of compounds identified in hydrochars (Ph, R-Ph, Su, and R-Su). BD = benzene derivative; HN = heterocyclic nitrogen-containing compounds; ES = aliphatic esters; CC = cyclic compounds; HS = heterocyclic sulfur-containing compounds; PH = phenolic compounds; HO = heterocyclic oxygen-containing compounds; PAH = polycyclic aromatic hydrocarbons; AC = alkanes, alkenes, alkynes; DA = dicarboxylic acids; SE = steroids; ST = sterols. The error bars represent standard deviation (SD).

**Figure 5 molecules-26-01026-f005:**
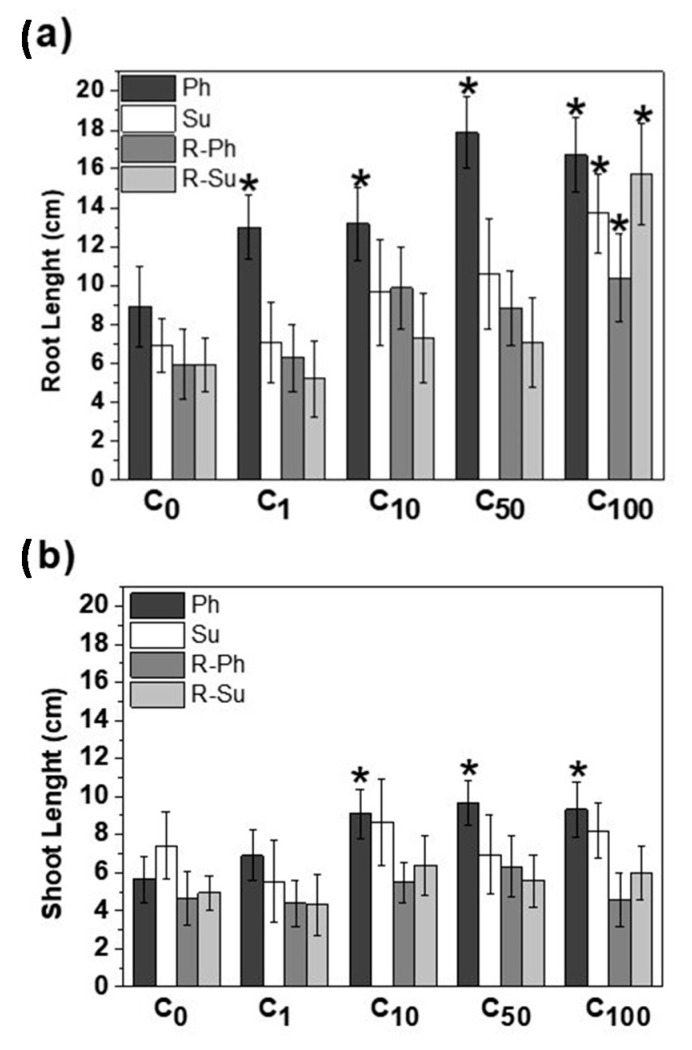
(**a**) Root and (**b**) shoot lengths of maize seeds after 7 days of growth in different OC concentrations of hydrochars (Ph, Su, R-Ph, and R-Su) soluble fractions. C_0_ = 0 mg C L^−1^, C_1_ = 1 mg C L^−1^, C_10_ = 10 mg C L^−1^, C_50_ = 50 mg C L^−1^, C_100_ = 100 mg C L^−1^. The error bars represent standard deviation (SD). (*) Indicates a positive statistical difference in relation to the control.

**Table 1 molecules-26-01026-t001:** Relative organic carbon (OC) distribution (%) over chemical shift regions (ppm) in 13C-CPMAS-NMR spectra of hydrochar samples and related structural index.

	^13^C-NMR Region (ppm)						
SAMPLE	0–43 Alkyl C	43–60 OCH_3_/NCH	60–90 O-alkyl-C	90–110 O-C-O Anomeric	110–145 Aromatic C-C/C-H	145–160 Aromatic C-O	160–190 COO/N-C=O
**Ph**	37.7	9.1	4.9	5.3	29.4	8.3	5.3
**R-Ph**	32.1	10.3	5.8	6.1	32.4	9.3	4.2
**Su**	42.9	7.3	4.3	4.7	28.3	7.3	5.2
**R-Su**	27.1	5.4	6.5	8.7	45.5	4.1	2.7
	**Structural Index**						
**SAMPLE**	***HB*/*HI***	***A*/*AO***	***ARM***	***LR***			
**Ph**	3.1	3.7	0.8	1.1			
**R-Ph**	2.8	2.7	1.0	1.1			
**Su**	3.7	4.8	0.7	1.0			
**R-Su**	3.3	1.8	1.2	1.3			

*HB*/*HI* = hydrophobicity index = [Σ (0–43) + (110–160)/Σ(43–60) + (60–110) + (160–190)]; *A*/*AO* = alkyl ratio = [(0–45)/(60–110)]; *ARM* = aromaticity index = [(110–160)/Σ(0–45) + (60–110)]; *LR* = Lignin Ratio= [(0–45)/(60–110)].

**Table 2 molecules-26-01026-t002:** Parameters and starting materials of hydrothermal carbonization reactions.

Sample	Ph	Su	R-Ph	R-Su
(Reaction)	(1)	(2)	(3)	(4)
Biomass	Bagasse	Bagasse	Bagasse	Bagasse
Liquid	Vinasse	Vinasse	Process Water from reaction 1	Process Water from reaction 2
Biomass/Liquid ratio	1:20	1:20	1:20	1:20
Temperature	230 °C	230 °C	230 °C	230 °C
Reaction time	13 h	13 h	13 h	13 h
Additive	Phosphoric acid (4% *v*/*v*)	Sulfuric acid (4% *v*/*v*)	Phosphoric acid (4% *v*/*v*)	Sulfuric acid (4% *v*/*v*)

## Data Availability

The data presented in this study are available in this article.

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
