# Peer review of "Insights on Molecular Characteristics of Hydrochars by 13C-NMR and Off-Line TMAH-GC/MS and Assessment of Their Potential Use as Plant Growth Promoters"

_molecules, 2021, doi:10.3390/molecules26041026_

Round 1

Reviewer 1 Report

The manuscript titled ‘Insights on molecular characteristics of hydrochars by 13C NMR and Off-line TMAH-GC/MS and assessment of their potential use as plant growth promoters’ analyzed different hydrochars’ chemical compositions prepared by different techniques and their effect on maize seed germination.

The topic of the manuscript is interesting for the Molecules’ readership, as hydrochar chemical compositions were evaluated using different instruments. While the paper explained many aspects of the differences between the treated hydrochars, it would need some improvement, especially in the Materials and Methods and the Results section (by adding the statistical data) to further enhance the paper’s merit.

In general, the paper requires some revisions before considering publication.

I have added some additional comments to the paper, collected below.

Specific comments:

General comments: the paper should be checked for typos, there are many in the pdf version.

Lines 98-101. This paragraph is better in the Materials and Methods section than in the results.

Figure 1. Statistical differences should be added to the graphs. All figure titles also should include all necessary information to stand alone in the paper, such as the abbreviations used in the figure, the number of replicates, the error bars if SD or SE, etc.

Lines 113-120. Maybe these groups would be better to show in a Table format.

Figure 2. The limits of the 7 groups could be shown in the figure by vertical lines.

Line 147. Were these differences statistically evaluated? Were any of them significant?

Line 254. Please include 145-160 ppm.

Table 1. The statistical differences should be included by using small letters.

Figure 4. The standard deviation should be included in the graphs.

Line 208. How significant the correlation was? Do you have any p and r values to back up this statement? Please amend.

Lines 218-225. P values should be included when significance is mentioned.

Line 228. P < 0,05 should be 0.05.

Figure 5. It is better to use * (or ** and ***) instead of a letter if only one difference is illustrated. However, in most cases, the letters are preferred, but using for all treatments/groups.

Lines 297-306. What was the amount of hydrochar generated per treatment?

How were the replications performed? How many replicates were per treatment?

Lines 298-299. What is the difference between reactions 1 and 2? Is there any? Please explain it in more detail.

Line 318. Air-dried?

Line 371. Zea mays should be in italics.

Lines 394-396. Was this statement proven by this study? Or it is the most likely scenario based on literature? Please amend the measurements in the Materials and methods section or clarify your statement.

Author Response

We are grateful to the Reviewers for the careful evaluation of our manuscript and the valuable indication to improve the scientific appraisal; please find enclosed in the attached file the detailed list of response to the Reviewer's comments

Reviewer 2 Report

 This paper assessed the potential of hydrochars of biomass use as plant growth promoters. The following problems need to be improved:

  1. The innovation of this chapter should be better highlighted in the Introduction section.
  2. The reason for adding sulfuric or phosphoric acids in the hydrothermal carbonization process should be explained.
  3. The diagram should be self-explanatory. The meaning of C should be given in Fig.S1.
  4. Please give the degree of matching of each compound by thermochemolysis- GC/MS.
  5. The calculation method of germination index should be given.
  6. Line 298-299, the meaning of reactions 1-4 are unclear and should be further stated.
  7. In conclusion, the most suitable hydrochars preparation process parameters for seed germination and the recommended concentration should be given.
  8. In terms of the structure of the paper, it is suggested to place the section of "Materials and Methods" before the section of "Results and Discussion".

Author Response

(The authors gave the same response as above.)

Round 2

Reviewer 1 Report

The authors revised the manuscript in accordance with the written suggestions. 

Reviewer 2 Report

The questions the last version mentioned have been solved, I think the paper can be accepted as it is.